# Human Salivary Histatin-1 Attenuates Osteoarthritis through Promoting M1/M2 Macrophage Transition

**DOI:** 10.3390/pharmaceutics15041272

**Published:** 2023-04-18

**Authors:** Antong Wu, Janak Lal. Pathak, Xingyang Li, Wei Cao, Wenchao Zhong, Mingjing Zhu, Qiuyu Wu, Wanyi Chen, Qiao Han, Siqing Jiang, Yuzhuo Hei, Ziyi Zhang, Gang Wu, Qingbin Zhang

**Affiliations:** 1Department of Temporomandibular Joint, Affiliated Stomatology Hospital of Guangzhou Medical University, Guangzhou Medical University, Guangzhou 510182, China; 2Guangdong Engineering Research Center of Oral Restoration and Reconstruction, Guangzhou Key Laboratory of Basic and Applied Research of Oral Regenerative Medicine, Guangzhou 510182, China; 3Department of Oral and Maxillofacial Surgery/Pathology, Amsterdam UMC and Academic Center for Dentistry Amsterdam (ACTA), Vrije Universiteit Amsterdam (VU), Amsterdam Movement Science (AMS), 1081 LA Amsterdam, The Netherlands; 4Department of Oral Cell Biology, Academic Centre for Dentistry Amsterdam (ACTA), University of Amsterdam and Vrije Universiteit Amsterdam, Amsterdam Movement Sciences, 1081 LA Amsterdam, The Netherlands

**Keywords:** histatin-1, macrophages, chondrogenic lineage cells, osteoarthritis, synovium

## Abstract

Osteoarthritis (OA) is an inflammation-driven degenerative joint disease. Human salivary peptide histatin-1 (Hst1) shows pro-healing and immunomodulatory properties. but its role in OA treatment is not fully understood. In this study, we investigated the efficacy of Hst1 in the inflammation modulation-mediated attenuation of bone and cartilage damage in OA. Hst1 was intra-articularly injected into a rat knee joint in a monosodium iodoacetate (MIA)-induced OA model. Micro-CT, histological, and immunohistochemical analyses showed that Hst1 significantly attenuates cartilage and bone deconstruction as well as macrophage infiltration. In the lipopolysaccharide-induced air pouch model, Hst1 significantly reduced inflammatory cell infiltration and inflammation. Enzyme-linked immunosorbent assay (ELISA), RT-qPCR, Western blot, immunofluorescence staining, flow cytometry (FCM), metabolic energy analysis, and high-throughput gene sequencing showed that Hst1 significantly triggers M1-to-M2 macrophage phenotype switching, during which it significantly downregulated nuclear factor kappa-B (NF-κB) and mitogen-activated protein kinases (MAPK) signaling pathways. Furthermore, cell migration assay, Alcian blue, Safranin O staining, RT-qPCR, Western blot, and FCM showed that Hst1 not only attenuates M1-macrophage-CM-induced apoptosis and matrix metalloproteinase expression in chondrogenic cells, but it also restores their metabolic activity, migration, and chondrogenic differentiation. These findings show the promising potential of Hst1 in treating OA.

## 1. Introduction

Osteoarthritis (OA) is the most prevalent degenerative joint disease, affecting nearly 7% of the world’s population [1,2]. OA exhibits articular cartilage/subchondral bone deconstruction, osteophyte formation, joint capsule hypertrophy, and synovial lining inflammation, leading to disability and impaired quality of life [3]. The etiology of OA may be multifactorial, including both systemic factors (e.g., genetics, changes in estrogen level, dietary intake, and bone mineral density) or mechanical factors (e.g., heavy body weight, muscle weakness, and joint stiffness) [1,4,5]. In recent years, low-grade inflammation was shown to play a vital role in the onset and progression of OA. During the early stages of OA, immune cells such as macrophages accumulate in the intimal lining layer, resulting in synovitis [6,7]. Thereafter, macrophages polarize into the pro-inflammatory M1 phenotype [8,9] and release high levels of reactive oxygen species (ROS), interleukin (IL)-1β, inducible nitric oxide synthase (iNOS), tumor necrosis factor-α (TNF-α), and IL-6 to promote inflammation [3,10,11], thereby causing a series of pathological changes, such as the expression of matrix metalloproteinases (MMPs), extracellular matrix (ECM) degradation, decreased chondrocyte autophagy, and enhanced apoptosis [12,13,14,15]. Therefore, anti-inflammation therapy has become a hot topic in the field of OA treatment.

For example, non-steroidal anti-inflammatory drugs (NSAIDs) and analgesic drugs are applied in the clinic as adjunctive therapy to physical therapy [16]. Furthermore, the intra-articular injection of hyaluronic acid (HA) may smoothen joint lubrication and mildly inhibit inflammation [17]. However, neither the combination of physical therapy and systemic medication nor HA injection can completely prevent the progression of OA [18,19]. Although some inhibitors of pro-inflammatory cytokines (e.g., anakinra, an IL-1 receptor antagonist, and AMG108, an IL-1 receptor type I monoclonal antibody) have been tested in clinical trials, hitherto, no satisfactory results have been achieved [20]. Hence, it is very important to find a safe and effective immunomodulatory drug with tissue repair and regeneration functions in the early intervention of OA.

As a promising candidate, histatin-1 (Hst1), a low molecular peptide containing 38 amino acids secreted mainly by human salivary glands, exhibits potent cell-activating functions including migration, adhesion, and differentiation [21,22,23,24,25]. Our recent study shows that Hst1-loaded hydrogel may efficiently repair critical-size osteochondral defects in temporomandibular joints [26]. In an acute skin wound model, we showed that Hst1 not only promotes wound healing but also inhibits the expression of IL-1β, IL-6, TNF-α, and C-reactive protein (CRP) [27]. Furthermore, a recent in vitro study shows that Hst1 inhibits the expression of pro-inflammatory cytokines in LPS-treated macrophages [28]. Considering its potent pro-healing and anti-inflammatory effects, Hst1 may be a promising drug to alleviate cartilage and bone deconstruction and prevent OA progression.

In this study, we investigated the efficacy of Hst1 in attenuating bone and cartilage damage in OA. To reveal its underlying mechanisms, we assessed the effect of Hst1 on the M1-to-M2 phenotype switch of macrophages and ascertain whether the Hst1-attenuated inflammatory response of macrophages could lead to the attenuation of inflammation-induced chondrogenic cell apoptosis and dysfunction.

## 2. Materials and Methods

### 2.1. Animal Study

Five-week-old male SD rats (180–225 g) and nine-week-old C57/BL6 male mice (18–22 g) were obtained from Southern Medical University (Guangzhou, China). After arrival at the animal care facility, animals were permitted at least 1 week to acclimate to the new environment and were provided with standard laboratory chow and water. All protocols for the animal experiments were approved by the Guangdong HUA WEI Testing Co., Ltd., Guangzhou, China (No. 20210601 and No. 20210801).

### 2.2. OA Animal Model

This study used a monosodium iodoacetate (MIA)-induced rat knee OA model. Eighteen SD rats were randomly divided into three groups: control, MIA, and MIA+Hst1. They were anesthetized with 2% sodium pentobarbital (40 mg/kg body weight). MIA (3 mg) dissolved in 50 μL phosphate buffer solution (PBS) was injected into the rat knee articular cavity. After 2 weeks, MIA group mice were injected with PBS (50 μL), and MIA+Hst1 group mice were injected with Hst1 (40 μM, 50 uL) twice a week for 4 weeks. The rats were euthanized via rapid and excessive intraperitoneal injection of sodium pentobarbital. Studies from the literature used the range of 5–100 μM of Hst1 for different in vitro and in vivo studies including wound healing, angiogenesis, and cartilage regeneration [26,29,30,31]. Therefore, we chose a tentative middle concentration of 40 μM to treat rat OA.

### 2.3. Micro-CT Analysis

Rat joints were harvested and fixed with 4% paraformaldehyde (PFA). Specimens were scanned using a Micro-CT scanner (μCT, SkyScan, Aartselaar, Belgium) set at a resolution of 10 μm with a voltage of 60 kV and an electric current of 100 μA. Pre-fixed joints were scanned. The region of interest was defined to cover the whole subchondral bone in the tibial plateau. Three-dimensional structural parameters were analyzed, including bone volume/total tissue volume (BV/TV), bone volume (BV), trabecular number (Tb.N), and trabecular separation (Tb.Sp).

### 2.4. Histological Observation

Specimen decalcification was performed in 10% Ethylene Diamine Tetraacetic Acid (EDTA) for 12 weeks. Fixed skin tissues and decalcified joints were embedded in paraffin, sectioned with a rotary microtome into 4 μm thick serial sections (ThermoFisher, Waltham, MA, USA), and stained with Hematoxylin and Eosin (H&E) and Safranin O-Fast Green (S&F) (Solarbio, Beijing, China). Immunohistochemistry staining was accomplished using anti-Aggrecan rabbit pAb (GB11373, Servicebio, Wuhan, China), anti-rabbit F4/80 pAb (GB11027, Servicebio, Wuhan, China), and anti-rabbit CD163 pAb (ab182422, Abcam, Cambridge, UK). In brief, after paraffin sections were deparaffinized, antigens were retrieved using citrate antigen retrieval buffer (Servicebio, China), and endogenous peroxidase activity was blocked with 3% hydrogen peroxide (Disinfection Technology, Hong Kong, China) and with 3% BSA (Servicebio, China). Sections were incubated with primary antibody overnight at 4 °C and then incubated with rabbit secondary antibody (HRP labeled, Servicebio, China) for 50 min at room temperature. Finally, the newly prepared DAB (Servicebio, China) solution was added, the color development time was controlled under the microscope, and the positive was brownish yellow. Images were captured using a light microscope (Leica Microsystems, Wetzlar, Germany). The positive staining area of each specimen was calculated with Image J software and measured on three consecutive specimens in each group. At least 3 specimens were measured in each group.

### 2.5. Air Pouch Acute Inflammation Model

The mouse skin air pouch acute inflammation model was established as described previously [32]. Briefly, twenty-four C57/BL6 mice were randomly divided into three groups: control, lipopolysaccharides (LPS), and LPS+Hst1. Chloral hydrate (4%) (10 mg/kg body weight) was used as an anesthetic agent. Sterile air (5 mL) was injected into the skin on the back of the mice. Three days later, sterile air (3 mL) was injected into the same cavity. On day 7, 1 mL PBS was injected in the control group, and 1 μg/mL LPS (Sigma Aldrich, St. Louis, MI, USA) was injected in the LPS group and the LPS+Hst1 group. After 1 h of PBS/LPS injection, Hst1 (40 μM, 1 mL) was injected in the LPS+Hst1 group, and 1 mL PBS was injected in the control group and LPS group. After 5 h, 0.9% saline solution (1 mL) was used to lavage the exudates from the air pouch. The supernatants of the exudates were measured for TNF-α, IL-1β, and IL-6 levels using mouse TNF-α, IL-1β, and IL-6 ELISA Kits. The mice were euthanized, and skin tissues were harvested for histology.

### 2.6. Cell Culture

The mouse cell line RAW264.7 (National Collection of Authenticated Cell Cultures, Shanghai, China) was used as a model of macrophages. The RAW264.7 macrophages were cultivated in high-glucose DMEM (Gibco, New York, NY, USA) containing 10% fetal bovine serum (FBS, Gibco, USA) and 1% Penicillin–Streptomycin solution (Gibco, USA) under 37 °C and in a humidified 5% CO_2_ incubator (ThermoFisher, USA). The chondrogenic cell line ATDC5 (National Collection of Authenticated Cell Cultures, China) was cultivated in DMEM/F12 (Gibco, USA) containing 10% FBS, 1% Penicillin-Streptomycin solution, and 1% ITS (Sigma Aldrich, USA). Both RAW264.7 macrophages and ATDC5 cells in 80% confluency were used for subsequent in vitro studies.

### 2.7. Effects of Hst1 on Cell Metabolic Activity

The lyophilized Hst1 peptide with a purity of 95% was purchased from Shanghai Top-Peptide Biotechnology Co. Ltd. (DPSHEKRHHGYRRKFHEKHHSHREFPFYGDYGSNYLYDN). ATDC5 cells (5000 cells/well) or macrophages (3000 cells/well) were cultured in 96-well culture plates in the presence of 0, 5, 10, 20, or 40 μM Hst1. Cell metabolic activity was evaluated using the cell counting kit-8 assay (CCK8, Dojindo, Kumamoto, Japan) [33].

### 2.8. LPS and Hst1 Treatment

RAW264.7 macrophages (3 × 10^5^ cells/well) were seeded in 6-well culture plates. After 80% confluency, cultures were divided into 3 groups: control, LPS, and LPS+Hst1. The control group only contains a cell culture medium, the LPS group was treated with *E. coli* LPS (100 ng/mL, Sigma Aldrich, USA) for 24 h, and the LPS+Hst1 group was pretreated with LPS (100 ng/mL) for 24 h. The LPS-containing medium was discarded and then treated with Hst1(40 μM) for 24 h.

### 2.9. Cytokine Expression Analysis with ELISA

After the three groups of cells were treated as described previously, the culture supernatant or air pouch exudate was collected, centrifuged at 3000 rpm (845 g) for 10 min, and stored at 80 °C for further experiments. Levels of IL-1β, TNF-α, and IL-6 were measured in the culture supernatant using a mouse ELISA kit for TNF-α, IL-1β, and IL-6 (Thermo Fisher, Waltham, MA, USA).

### 2.10. RT-qPCR

The total RNA was extracted from cells using an RNA Extraction Kit (Accurate Biotechnology, Danyang, China). Total RNA was reverse-transcribed to cDNA using a one-step RT-PCR kit (Accurate Biotechnology, China). RT-qPCR was performed using SYBR Green (Accurate Biotechnology, China) in an RT-PCR machine (Agilent Technologies, Santa Clara, CA, USA). Values were normalized to GAPDH mRNA levels. The gene primer sequences used for qRT-PCR are listed in Table 1.

### 2.11. Immunofluorescence (IF)

Macrophages cultured in coverslip were fixed with 4% PFA for 15 min. Nonspecific binding was blocked using 1% bovine serum albumin (BSA, Solarbio, China). Then, samples were incubated with anti-iNOS (Cell Signaling Technology, Danvers, MA, USA) and anti-CD206 (Cell Signaling Technology, USA) primary antibodies overnight. The samples were incubated with Alexa Fluor 647-conjugated goat–anti-rabbit IgG (Beyotime, Haimen, China) for 1 h at room temperature. Further, cells were stained in 4, 6-diamidino-2 phenylindole (DAPI, Beyotime, China) for 5 min. Cells were imaged using a confocal fluorescence microscope (Leica Microsystems, Wetzlar, Germany).

### 2.12. Flow Cytometry (FCM)

An annexin V-FITC/propidium iodide double staining kit (Beyotime, China) was used for chondrogenic cell apoptosis rate analysis. Anti-CD86 (Biolegend, San Diego, CA, USA) was used to evaluate macrophage subsets. Briefly, RAW264.7 macrophages were collected from 6-well plates and washed three times with PBS. The cells (1 × 10^6^) were incubated in PBS with anti-CD86 at 4 °C for 30 min in the dark. Cells were washed twice, suspended in 500 μL PBS, and detected using flow cytometry (BD Biosciences, San Diego, CA, USA).

### 2.13. Metabolic Assays

A Seahorse Bioscience XF extracellular flux analyzer (Agilent Technologies, Santa Clara, CA, USA) measured the oxygen consumption rate (OCR). RAW264.7 macrophages were seeded in XF cell culture plates with Seahorse XF DMEM supplemented with 10 mM glucose, 2 mM glutamine, and 1 mM sodium pyruvate. The basal and maximal respiration, ATP production, and spare respiration capacity were measured as described previously [34].

### 2.14. Western Blot Assay

The cell lysate was prepared in RIPA lysis buffer (Beyotime, China) containing 1% protease inhibitors (PMSF, Beyotime, China) and 2% phosphatase inhibitor cocktail (Beyotime, China). The protein concentration in lysate was determined using a BCA protein assay kit (Beyotime, China). Samples were separated with SDS-PAGE for 90 min and blotted onto polyvinylidene difluoride (PVDF) membranes (Millipore, Burlington, MA, USA) for 30 min. The PVDF membranes were blocked with 5% (*w/v*) non-fat dried milk in TBST. Specific primary antibodies and horseradish peroxidase-conjugated secondary antibodies were used to detect the targeted protein. An Ultra-sensitive ECL ChemiLuminescence kit (Beyotime, China) was used to develop the color of the stained protein band. The Image J software was used to quantify the band densities. The antibodies used for Western blot and immunofluorescence staining are listed in Table 2.

### 2.15. High-Throughput Generation Sequencing

The total RNA of RAW264.7 macrophages was extracted using a Trizol plus RNA Purification Kit (Thermo Fisher, USA). RNA degradation and contamination were monitored on a 1% agarose gel run. RNA concentration and purity were checked using a NanoPhotometer spectrophotometer (IMPLEN, CA, USA). RNA integrity was assessed using the RNA Nano 6000 Assay Kit of the Agilent Bioanalyzer 2100 system (Agilent Technologies, USA). Library construction, sequencing strategy, and RNA-seq data analysis were performed as described in our previous work [35].

### 2.16. Chondrogenic Differentiation Assay

#### 2.16.1. Conditioned Medium (CM) Collection

CM was collected from 3 groups of macrophage cultures, i.e., Ctrl-CM (from macrophages), LPS-CM (from LPS-treated macrophages), and LPS+Hst1-CM (from LPS and Hst1-treated macrophages). LPS treatment was used to induce the M1 macrophage phenotype. The original medium containing LPS or Hst1 was discarded and replaced with a fresh serum-free medium 8 h before the collection of CM. The CM was centrifuged at 3000 rpm (845 g) for 10 min and kept at −80 °C. The CM was mixed with a fresh ATDC5 culture medium containing 20% FBS and 2% ITS at a ratio of 1:1 and used for the subsequent experiments.

#### 2.16.2. Effects of CM on Cell Metabolic Activity

ATDC5 cells (5000 cells/well) were cultured in 96-well plates in the presence of Ctrl-CM, LPS-CM, and LPS+Hst1-CM and incubated for 24, 48, and 72 h. Cell metabolic activity was evaluated using a CCK8 (Dojindo, Japan) kit.

#### 2.16.3. Cell Migration Assay

The chemotactic motility of chondrogenic cells was determined using a modified Boyden chamber assay with polycarbonate filters (8 μm pore size, ThermoFisher, USA). In brief, ATDC5 cells (1 × 10^4^ cells/well) suspended in 100 μL of DMEM/F12 containing 0.5% FBS were seeded in the upper chambers. The lower chambers were filled with 500 μL CM. Cells were allowed to migrate for 6 h. Non-migrated cells were removed with cotton swabs, and migrated cells were fixed with cold 4% PFA for 15 min and stained with 0.1% crystal violet for 15 min. Images were taken using a light microscope (Leica Microsystems, Germany), and migrated cells were quantified using Image J software.

#### 2.16.4. Alcian Blue Staining and Safranin O Staining

ATDC5 cells (3 × 10^4^ cells/well) seeded in 48-well plates were cultured with Hst1 or macrophage CM for 7 days. Cells were fixed with 4% PFA for 15 min and stained with Alcian blue (Solarbio, China) and Safranin O. Images were taken using a light microscope (Leica Microsystems, Germany).

### 2.17. Statistical Analysis

All of the data are expressed as means ± standard deviation (SD), and the significance of experimental data was analyzed using T-Tests and ANOVA followed by the Bonferroni post hoc test. All data analysis was conducted with Graphpad Prism 8 (Graphpad Prism, San Diego, CA, USA). The statistical significance was defined as *p* < 0.05.

## 3. Results

### 3.1. Hst1 Attenuated the Bone and Cartilage Damage in MIA-Induced Knee OA

Figure 1A illustrates the procedure of OA development in rats and the Hst1 treatment. Local injection of MIA in rat joints prominently deteriorated the cartilage and subchondral bone, as indicated by the micro-CT images (Figure 1B). Interestingly, the coronal, sagittal, and transverse micro-CT images showed that the local injection of Hst1 prevented the MIA-induced deconstruction of cartilage and subchondral bone (Figure 1B). The quantification of subchondral bone parameters from the micro-CT data showed reduced BV, Tb.N, and BV/TV in the MIA group compared to the control group (Figure 1C). Hst1 treatment rescued the MIA-induced loss of BV, Tb.N, and BV/TV in subchondral bone. However, the Tb.Sp remained unchanged in the control, MIA, and Hst1-treated-MIA groups. H&E- and S&F-stained tissue sections of the OA joint clearly showed inflamed synovium with infiltrated inflammatory cells and deteriorated menisci. Moreover, the MIA group showed a loss of articular cartilage, calcified cartilage, and subchondral bone. Hst1-treated OA joint histology showed better joint anatomy with less inflamed synovia and menisci, more intact articular and calcified cartilage, and less deteriorated subchondral bone (Figure 1D). Aggrecan expression was diminished in the joint cartilage of the MIA group and was recovered in Hst1-treated joints (Figure 1E). Similarly, macrophage infiltration was dramatically higher in OA joints, and Hst1 treatment reduced this phenomenon (Figure 2A,C). The number of CD163 expressing M2 macrophages was higher in the Hst1-treated-MIA group compared with the MIA group by 3.35-fold (Figure 2B,D). These results indicate the protective effect of Hst1 against MIA-induced osteochondral damage possibly via macrophage immunomodulation.

### 3.2. Hst1 Did Not Affect the Differentiation of Chondrogenic Lineage Cells

Hst1 (10, 20, and 40 µM) slightly induced the proliferation of ADTC5 cells but did not affect chondrogenic differentiation (Appendix A). These results indicate that the Hst1-mediated alleviation of OA was not via a direct effect on chondrogenic cell functions.

### 3.3. Hst1 Suppressed Inflammation in Acute Air Pouch Model in Mice

Because Hst1 mitigated the number of macrophages in OA joints, we further investigated the anti-inflammatory properties of Hst1 in vivo. The mice air pouch model was used to assess the anti-inflammatory properties of Hst1 (Figure 3A). The air pouch exudates from the LPS group showed remarkably higher levels of pro-inflammatory markers IL-1β, IL-6, and TNF-α (Figure 3B). Hst1 dramatically reduced the LPS-induced levels of pro-inflammatory markers (Figure 3B). Thicker and inflamed air pouch tissue with an increased number of inflammatory cells was observed in the LPS-treated group (Figure 3C). Hst1 mitigated these effects of LPS in air pouch tissue (Figure 3C,D). These results show that Hst1 has the potential to inhibit the infiltration of inflammatory cells and inflammation. OA is an inflammatory disease of joints, and inflammatory macrophages play a vital role in the progression of pathogenicity [36]. Based on these facts, we hypothesized that Hst1 mitigates OA progression possibly via anti-inflammatory effects.

### 3.4. Hst1 Mediated M1-to-M2 Macrophage Phenotype Transition

Hst1 at concentrations of 5 and 10 μM did not affect RAW264.7 macrophage metabolic activity, but 20 and 40 μM induced it (Figure 4A). LPS treatment robustly induced an M1-like pro-inflammatory phenotype, as indicated by the higher expression of proinflammatory cytokines IL-1β, IL-6, and TNF-α (Figure 4B). Hst1 treatment dramatically mitigated the LPS-induced expression of proinflammatory markers, and the dose of 40 μM Hst1 showed the most robust effect (Figure 4B). Based on these results, this study chose 40 μM Hst1 as an optimal dose of treatment for further studies. M1 macrophages produce nitric oxide (NO) to exert inflammatory responses [37]. Hst1 alleviated the LPS-induced NO level in macrophages (Figure 4C). We further analyzed the mRNA expression of M1 and M2 macrophage markers in LPS±Hst1-treated macrophages. Hst1 significantly reduced the LPS-induced expression of M1 phenotype markers IL-1β, IL-6, TNF-α, iNOS, and CD86 (Figure 4D). Interestingly, Hst1 robustly induced M2 phenotype markers CD206, IL-10, and Arg-1 in LPS-treated macrophages (Figure 4D). As confirmed using Western blot analysis, Hst1 inhibited CD86 and promoted CD206 expression in LPS-treated macrophages (Figure 4E). These results indicate the M1-to-M2 macrophage phenotype transition potential of Hst1. IF staining and FCM analysis further confirmed the M1-to-M2 macrophage phenotype transition potential of Hst1 (Figure 5A,B). In addition, compared with LPS treatment, Hst1 pretreatment showed a higher OCR (Figure 5C) and higher ATP production in RAW264.7 macrophages.

### 3.5. Hst1 Treatment Downregulated NF-κB and MAPK Signaling in LPS-Treated Macrophages

mRNA sequencing data showed 2231 differentially upregulated and 1508 differentially downregulated mRNAs in LPS-treated RAW264.7 macrophages compared to the control group (Figure 6A). A total of 693 differentially upregulated and 522 differentially downregulated mRNAs were detected in LPS+Hst1-treated RAW264.7 macrophages compared to the LPS-treated group (Figure 6B). Figure 6C shows 417 mRNA intersections of LPS vs. control upregulated genes and LPS+Hst1 vs. LPS downregulated genes. Furthermore, the results of the GSEA pathway enrichment analysis show that the NF-κB, ERK1/2, and MAPK signaling pathways were positively enriched in LPS-treated macrophages compared to the control group, and these signaling pathways were negatively enriched in Hst1+LPS-treated cells compared to those in the LPS-treated group (Figure 6D,E). Western blot analysis revealed that LPS markedly activated the phosphorylation of IκBα and NF-κB in macrophages, whereas Hst1 inhibited the LPS-induced phosphorylation of IκBα and NF-κB (Figure 7A). Similarly, LPS induced the phosphorylation of ERK, JNK, and P38, whereas Hst1 inhibited LPS-induced pJNK, pErk, and pP38 phosphorylation in M1 macrophages (Figure 7B). These results are consistent with the analysis of the mRNA sequencing results. Our results indicate the Hst1-mediated downregulation of MAPK and NF-κB signaling as a possible mechanism of Hst1-mediated M1-to-M2 macrophage phenotype transition.

### 3.6. Hst1 Restored Metabolic Activity, Migration, Chondrogenic Differentiation, and Matrix Destruction in LPS/Macrophage-CM-Treated ADTC5 Cells

We investigated the effects of M0 macrophage-CM, LPS-treated macrophage (M1)-CM, and Hst1-treated M1 macrophage-CM. M1 macrophage-CM inhibited the proliferation of chondrogenic cells compared to M0 macrophage-CM (Figure 8A). Interestingly, this effect was restored by Hst1-treated M1 macrophage-CM. As indicated by Alcian blue and Safranin O staining, M1 macrophage-CM robustly inhibited the chondrogenic differentiation of precursor cells compared to M0 macrophage-CM (Figure 8C), whereas this effect was restored by Hst1-treated M1 macrophage-CM. Furthermore, M1 macrophage-CM drastically inhibited the migration of chondrogenic cells compared to M0 macrophage-CM, and this effect was reversed by Hst1-treated M1 macrophage-CM (Figure 8B). These results indicate that M1 macrophages in OA could inhibit chondrogenic cell functions, and Hst1 has the potential to reverse the inhibitory effects of the M1 macrophage on chondrogenic cell functions via M1-to-M2 macrophage phenotype transition.

We further analyzed the mRNA expression of chondrogenic differentiation markers Col2, Sox9, and Aggrecan as well as cartilage tissue-degrading inflammatory factors MMP-3, MMP-9, MMP-13, and Adamts5 in chondrogenic cells. M1 macrophage-CM inhibited the mRNA expression of chondrogenic differentiation markers at day 7 compared to M0 macrophage-CM, and Hst1-treated M1 macrophage-CM induced Sox9, Col2, and Aggrecan mRNA expression (Figure 9A). The results from Western blot analysis were similar to the results of RT-qPCR analysis (Figure 9B). Moreover, M1-CM robustly promoted the expression of MMPs (MMP-3, MMP-9, and MMP-13) and Adamts5 mRNA expression in chondrogenic cells compared with M0 macrophage-CM. This effect of M1 macrophage-CM was reversed by the Hst1-treated M1 macrophage-CM (Figure 9A). The Western blot results of MMP expression supported the results of RT-qPCR analysis (Figure 9C). These results indicate the robust anabolic effect of Hst1 on chondrogenic cell functions via macrophagic immunomodulation.

### 3.7. Hst1 Attenuated LPS/Macrophage-CM-Induced Apoptosis in ATDC5 Cells

We further analyzed the effect of M0 macrophage-CM, M1 macrophage-CM, and Hst1-treated M1 macrophage-CM in chondrogenic cell apoptosis. M1 macrophage-CM robustly promoted the expression of apoptotic markers Bax and Caspase 3 in chondrogenic cells at day 3 compared with M0 macrophage-CM (Figure 10A). Hst1-treated M1 macrophage-CM reduced apoptotic markers’ expression in chondrogenic cells compared with the M1 macrophage-CM. Furthermore, the Hst1-treated M1 macrophage-CM group showed reduced apoptosis of chondrogenic cells compared with M1 macrophage-CM (Figure 10B). Results of Western blot analysis of apoptotic markers’ expression also corroborated the findings from RT-qPCR analysis (Figure 10C). These results indicate that the M1 macrophage could induce chondrogenic cell apoptosis during OA and that Hst1 could reverse this effect via M1-to-M2 macrophage phenotype switching-regulated immunomodulation in OA joints.

## 4. Discussion

Hst1 has shown both cell-activating functions and immunomodulatory potential [38,39]. However, the protective effects of Hst1 against bone and cartilage degeneration in OA as well as the underlying mechanisms remain to be investigated. In this study, we, for the first time, showed that the articular injection of Hst1 alleviates rat knee OA as indicated by the intact cartilage, subchondral bone, and bone marrow structure as well as less inflamed synovial membranes and menisci. Macrophage infiltration was dramatically reduced in Hst1-treated OA joints. Hst1 did not alter chondrogenic differentiation but robustly inhibited inflammation. Hst1 switched from the M1 to the M2 phenotype via the inhibition of MAPK/NF-κB signaling in M1 macrophages. Hst1-initiated M1-to-M2 macrophage polarization remarkably inhibited inflammation. Hst1 rescued M1-macrophage-inhibited chondrogenic cell functions including metabolic activity, migration, and chondrogenic differentiation via M1-to-M2 macrophage polarization. Moreover, Hst1 inhibited M1 macrophage-induced apoptosis of chondrogenic cells. In summary, Hst1 alleviated rat knee OA progression via the M1-to-M2 macrophage phenotype switching-mediated mitigation of inflammation and anabolic effects on chondrogenic cells’ functions, suggesting the possible therapeutic potential of Hst1 to treat knee OA.

Our previous studies show that Hst1 can promote osteochondral regeneration in temporomandibular joint defects in rabbits [26]. In the current study, Hst1 alleviated MIA-induced OA in rat knee joints. Hst1 prevented OA-induced damage of cartilage and subchondral bone in the knee. In the early stages of OA, the cartilage surface shows hypertrophic changes and chondrocyte apoptosis. At the same time, chondrocyte hypertrophy-like changes produce matrix-degrading enzymes, including aggrecanases and MMPs, which can lead to protease-mediated degradation of the cartilage extracellular matrix and loss of proteoglycan [6,40]. The promotion of aggrecan production and mitigation of aggrecan degradation can prevent the destruction of joints in the early stage of OA [41]. In this study, OA dramatically degraded aggrecan in joints, and Hst1 seemed to robustly prevent aggrecan degradation in OA joints. These results indicate the protective effect of Hst1 on joint anatomical structure during OA.

Various growth factors were reported to directly promote chondrogenic cell functions and to accelerate osteochondral regeneration in OA joints [42,43,44,45]. The results of our in vitro study show no significant influence of Hst1 in the chondrogenic differentiation of precursor cells, suggesting that the protective effect of Hst1 during OA was not via a direct effect on chondrogenic cell functions. Hst1 has anti-inflammatory properties [28]. Inflammation levels in the OA joint are determined by the presence of immune cells including macrophages [46]. Macrophages are one of the most abundant immune cells in the synovium [9] and participate in various joint diseases, such as gouty arthritis [47], rheumatoid arthritis, and OA [48,49]. In the current study, the Hst1-treated OA joint showed a dramatically reduced number of macrophages compared with the untreated arthritic joints. CD163 is an M2 macrophage marker and was upregulated in Hst1-treated OA joints compared with OA joints, indicating the possible role of Hst1 in macrophage immunomodulation. Furthermore, Hst1 dramatically inhibited the levels of inflammatory markers, e.g., IL-1β, IL-6, and TNF-α, in the mouse skin inflammatory air pouch model. Moreover, Hst1 reduced the number of infiltrated inflammatory cells in inflamed skin subcutaneous tissue. These results allowed for speculation that the macrophage immunomodulation-mediated anti-inflammatory role of Hst1 possibly mitigates OA pathogenicity rather than having a direct effect on chondrogenic cell functions.

Macrophage-mediated immunomodulation plays a vital role in OA pathogenicity and cartilage repair [50]. In OA joints, macrophages mainly polarize to the M1 phenotype and release IL-1β, TNF-α, and IL-6, which creates the pro-inflammatory milieu [51]. M2 macrophages release IL-4, IL-10, and TGF-β to exert anti-inflammatory effects [52]. The M2 polarization of macrophages in OA joints was reported to alleviate inflammation and joint destruction as well as to promote cartilage and subchondral bone regeneration [49,53]. Therefore, the M1-to-M2 macrophage phenotype transition is a recent focus in the field of OA therapy [49]. An in vitro study reported that Hst1 mitigates M1 macrophage-induced inflammation [28]. However, the M1-to-M2 phenotype switching potential of Hst1 has not been investigated yet. We found that Hst1 effectively switched the M1 macrophages to M2, inhibiting the release of proinflammatory cytokines IL-1β, TNF-α, and IL-6 and promoting the release of anti-inflammatory cytokine IL-10. Oxygen consumption rate (OCR) and ATP production are higher in M2 macrophages compared to M1 macrophages [54]. Hst1-treated M1 macrophages showed higher OCR and ATP production, thereby confirming the M1-to-M2 phenotype switch. Our current study, for the first time, reports that Hst1 has the robust potential to switch M1-to-M2 macrophage polarization. The activation of NF-κB signaling dictates the M1 macrophage polarization and downregulation of NF-κB signaling in M1 macrophages, which triggers the phenotype switch to M2 macrophages [55]. This study reveals that Hst1 downregulates NF-κB and MAPK signaling in M1 macrophages and triggers their switch to M2 macrophages. These results suggest that Hst1 triggers M1-to-M2 macrophage polarization by downregulating NF-κB and MAPK signaling pathways.

Because Hst1 did not directly affect the chondrogenic differentiation of precursor cells but initiated M1-to-M2 macrophage polarization, we further analyzed the effect of Hst1-mediated M1-to-M2 macrophage polarization on chondrogenic cells’ functions. M1 macrophage-released pro-inflammatory cytokines such as IL-1β, TNF-α, and IL-6 inhibit matrix production and induce the production of cartilage matrix-degrading MMPs [56,57]. Moreover, the M2 macrophage was reported to promote cartilage repair in degenerative OA [58]. In this study, M1 macrophages inhibited the metabolic activity, migration, and chondrogenic differentiation of precursor cells. Interestingly, the Hst1-mediated M1-to-M2 phenotype switch of macrophages rescued the M1 macrophage-inhibited proliferation, migration, and chondrogenic differentiation of precursor cells. Pro-inflammatory cytokines released in arthritic joints promote MMP expression in chondrocytes [59]. We found that M1 macrophages robustly promoted the expression of MMP2, MMP3, MMP9, MMP13, and Adamts5 in chondrogenic cells. As expected, the Hst1-mediated M1-to-M2 phenotype switch of macrophages significantly attenuated M1 macrophage-induced MMP expression in chondrogenic cells. These results indicate that Hst1 induced M1-to-M2 macrophage polarization and has the potential to promote chondrogenic cell metabolic activity, migration, and chondrogenic differentiation. In this study, the Hst1-mediated M1-to-M2 macrophage phenotype switch robustly inhibited the M1 macrophage-induced apoptosis of chondrogenic cells. Overall, our results indicate that Hst1 exerts chondroprotective effects during OA via M1 to macrophage phenotype transition.

Although this study extensively analyzed the efficacy of Hst1 in the inflammation modulation-mediated attenuation of bone and cartilage damage in OA, some limitations do exist. The lack of a control group using scrambled peptides is a limitation of this study. The mechanism of the Hist-1-mediated effect on macrophage immune modulation should be further studied, with such a study mainly focusing on Hist-1 receptors in macrophages. Moreover, the possible systemic adverse effect of Hist-1 treatment should be thoroughly investigated during an in vivo study to evaluate its biological safety.

## 5. Conclusions

Our results show that Hst1 significantly promotes the macrophage M1-to-M2 transition to attenuate bone and cartilage deconstruction in OA possibly by downregulating the NF-κB and MAPK signaling pathways.

## Figures and Tables

**Figure 1 pharmaceutics-15-01272-f001:**
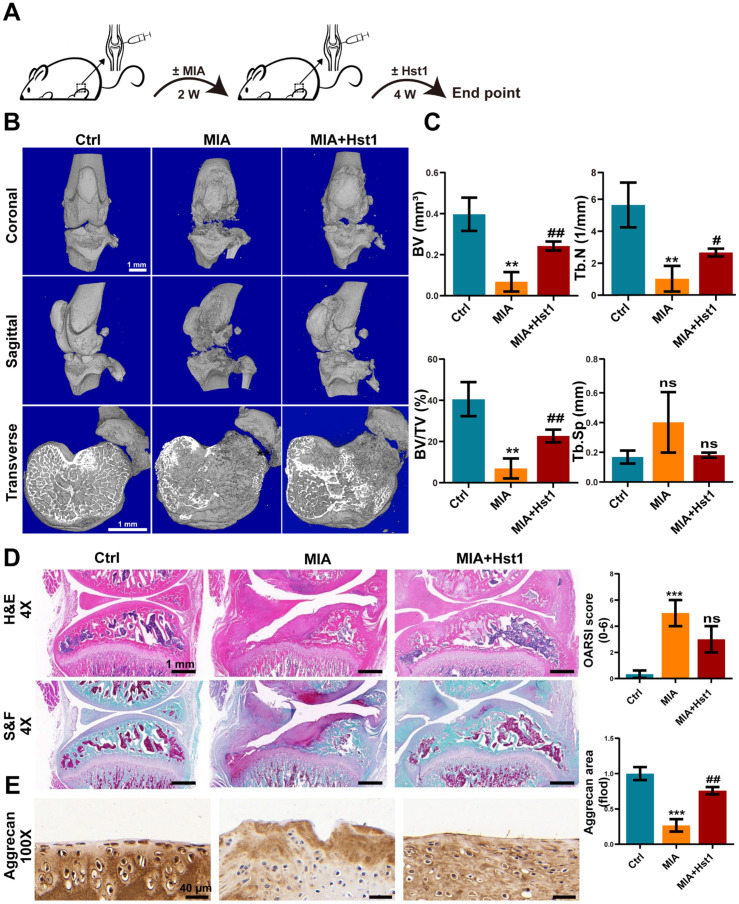
Hst1 attenuated bone and cartilage damage in MIA-induced knee OA. (**A**) The procedure of creating the rat knee OA model and administering Hst1 treatment. (**B**) Three-dimensional micro-CT images of the transverse section, coronal, and sagittal planes of the knee joint, scale bar = 1 mm. (**C**) Quantitative analysis of skeletal parameters BV, BV/TV, Tb. N, and Tb.Sp. (**D**) Knee joint tissue sections were stained with S&F and H&E, and OARSI was used for quantitative analysis, scale bar = 1 mm. (**E**) Representative images of Aggrecan immunohistochemical staining of knee joint tissue sections, scale bar = 40 μm. Data are expressed as means ± SD from three independent experiments. Significant differences compared to control group: ** *p* < 0.01 and *** *p* < 0.001, and compared to MIA group: # *p* < 0.05 and ## *p* < 0.01; ns: not significant.

**Figure 2 pharmaceutics-15-01272-f002:**
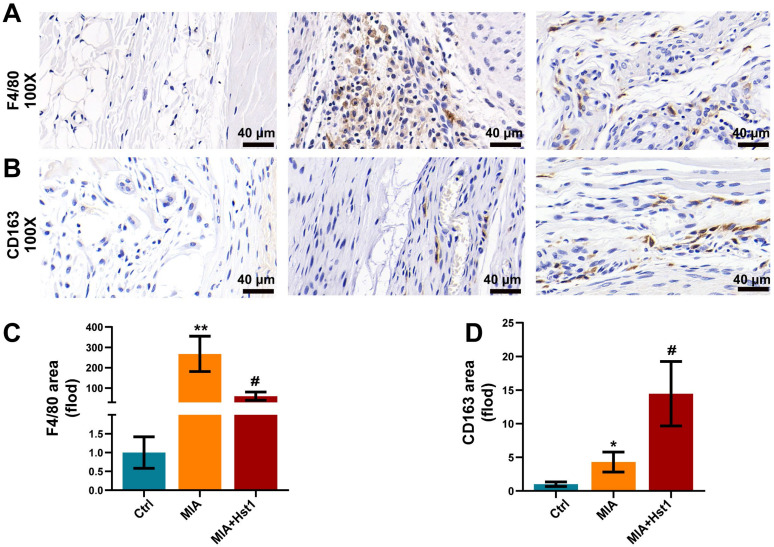
Hst1 reduced macrophage infiltration in the MIA-induced knee OA. (**A**,**B**) Synovial macrophages of the knee joint were identified by using F4/80 immunohistochemistry; moreover, CD163 was chosen for marking M2 macrophages. (**C**,**D**) Quantitative analysis of synovial macrophages and M2 macrophages, scale bar = 40 μm. Data are expressed as means ± SD from three independent experiments. Significant differences compared to the control group: * *p* < 0.05 and ** *p* < 0.01, and compared to MIA group: # *p* < 0.05.

**Figure 3 pharmaceutics-15-01272-f003:**
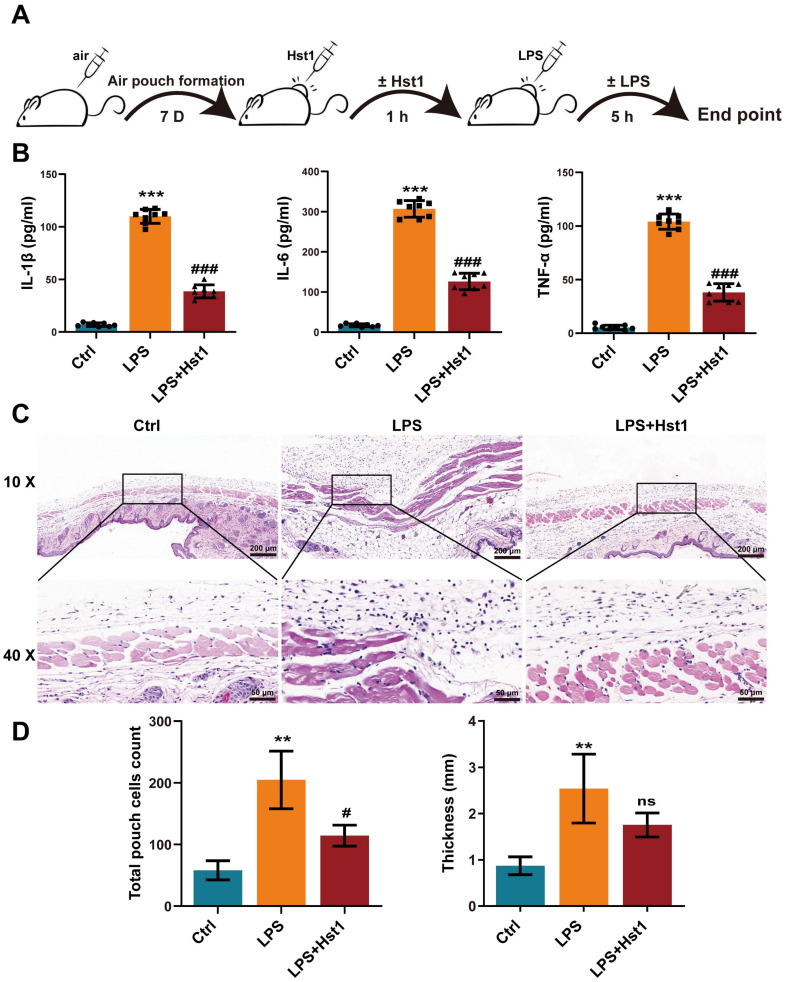
Hst1 suppressed inflammation in acute air pouch model in mice. (**A**) The procedure of creating the mice air pouch model and administering Hst1 treatment. (**B**) The production levels of IL-6, IL-1β, and TNF-α in the air pouch lavage fluid were determined by using ELISA, n = 8. (**C**) H&E staining of air pouch subcutaneous tissue sections, n = 3; scale bar = 200 μm in top row and 50 μm in bottom row. (**D**) Quantification of infiltrated cells and thickness of subcutaneous tissue, n = 3. Data are expressed as means ± SD from ≥ three independent experiments. Significant differences compared to control group: ** *p* <0.01 and *** *p* < 0.001, and compared to LPS group: # *p* <0.05 and ### *p* < 0.001; ns: not significant.

**Figure 4 pharmaceutics-15-01272-f004:**
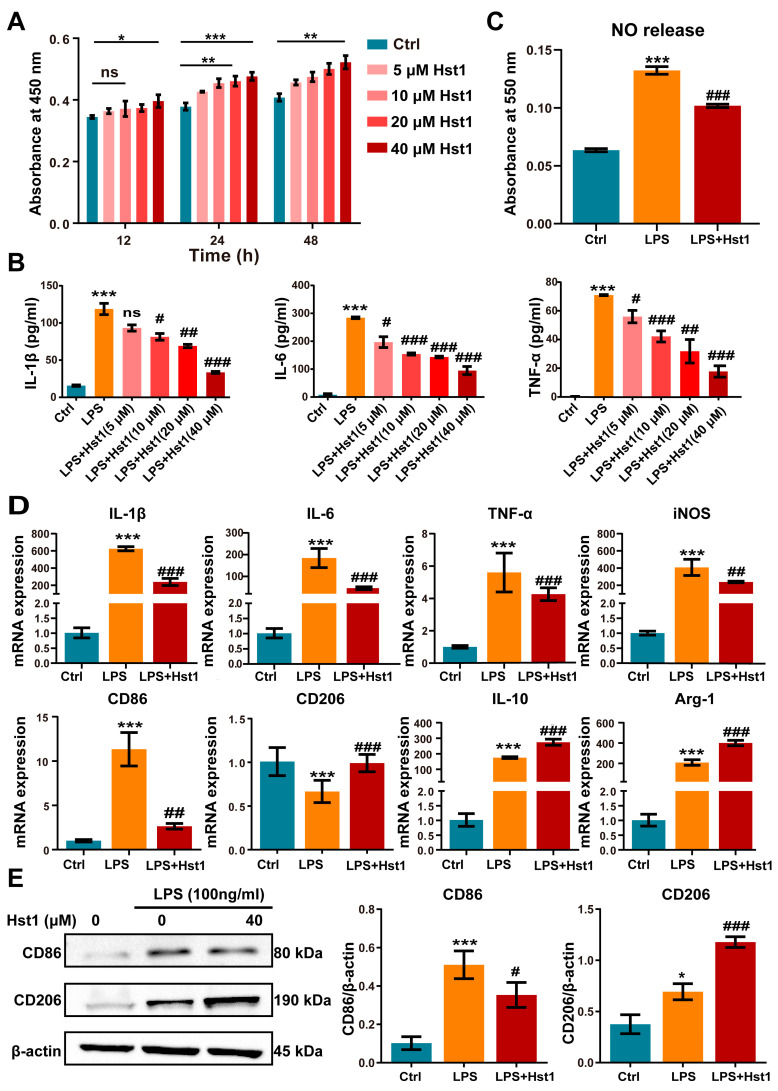
Hst1 promoted the M1-to-M2 macrophage phenotype transition. (**A**) CCK8 assay was used to analyze the metabolic activity of macrophages, n = 3. (**B**) Hst1-mediated mitigation of LPS-induced inflammatory cytokines IL-6, IL-1β, and TNF-α expression in macrophages determined by using ELISA, n = 3. (**C**) The production level of NO in LPS- and LPS+Hst1 (40 µM)-treated macrophages, n = 4. (**D**) mRNA level expressions of M1 macrophage markers IL-1β, IL-6, TNF-α, iNOS, and CD86 and of M2 macrophage markers IL-10, CD206, and Arg-1, n = 6. (**E**) Protein expressions of CD86 and CD206 in macrophages were analyzed by using Western blot, n = 3. Blots are cropped from Appendix A. Data are expressed as means ± SD from ≥three independent experiments. Significant differences compared to the control group: **p* < 0.05, ** *p* < 0.01, and *** *p* < 0.001, and compared to LPS group: # *p* < 0.05, ## *p* < 0.01, and ###: *p* < 0.001; ns: not significant.

**Figure 5 pharmaceutics-15-01272-f005:**
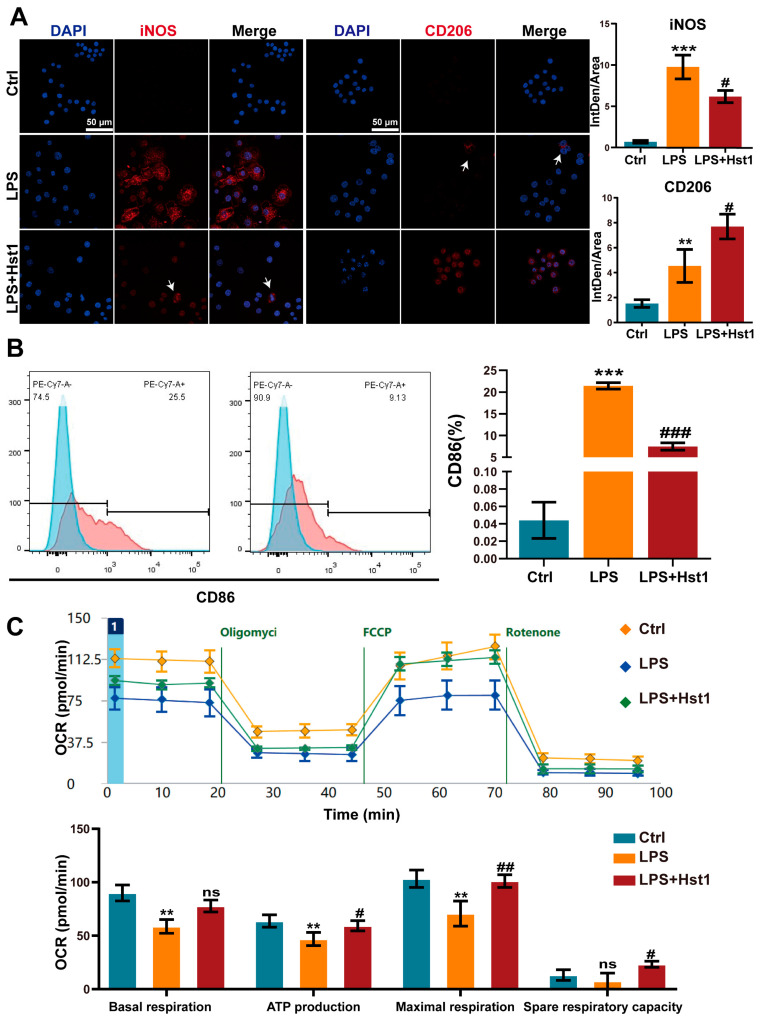
Effects of Hst1 on the polarization phenotype of macrophages. (**A**) The M1 macrophage marker iNOS and M2 macrophage marker CD206 were incubated with primary antibody and observed using confocal microscopy, n = 3; scale bar = 50 μm. (**B**) M1 macrophage subsets evaluated using flow cytometry and by staining CD86, n = 3. (**C**) Energy metabolism of macrophages analyzed using seahorse analysis, n = 4. Data are expressed as means ± SD from ≥ three independent experiments. Significant differences compared to the control group: ** *p* < 0.01, and *** *p* < 0.001, and compared to LPS group: # *p* < 0.05, ## *p* <0.01, and ### *p* < 0.001; ns: not significant.

**Figure 6 pharmaceutics-15-01272-f006:**
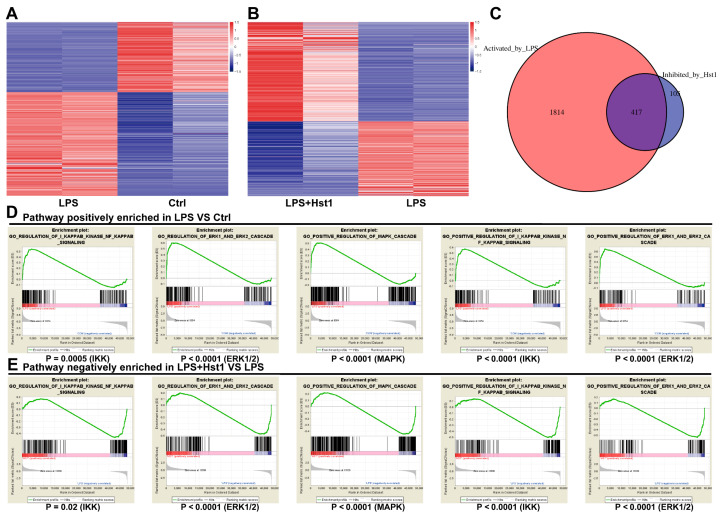
High-throughput gene sequencing of macrophages. (**A**,**B**) Differential mRNA expression of RAW264.7 macrophages in each group. (**C**) mRNA intersections of LPS vs. control upregulated genes and LPS+Hst1 vs. LPS downregulated genes. Gene set enrichment analysis (GSEA) shows enrichment of the NF-κB and MAPK pathways. (**D**) Pathways positively enriched in LPS group vs. control group and (**E**) pathways negatively enriched in LPS+Hst1 group vs. LPS group.

**Figure 7 pharmaceutics-15-01272-f007:**
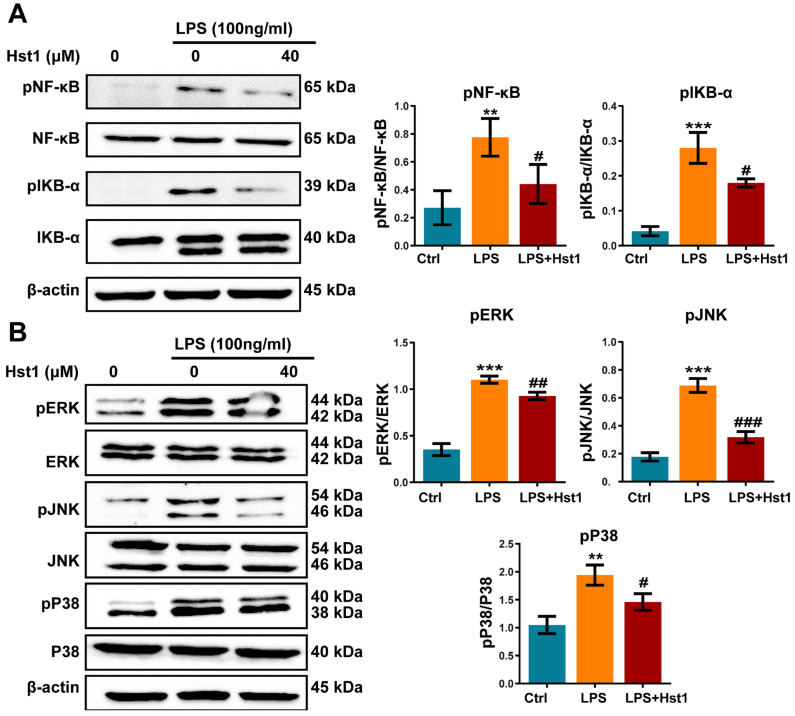
Hst1 treatment downregulated NF-κB and MAPK signaling in LPS-treated macrophages. (**A**,**B**) Western blot analysis and qualification of NF-κB signaling-related proteins MAPK signaling-related proteins pERK/ERK, pJNK/JNK, and pP38/P38. Blots are cropped from Appendix A. Data are expressed as means ± SD from three independent experiments. Significant differences compared to the control group: ** *p* < 0.01 and *** *p* < 0.001, and compared to LPS group: # *p* < 0.05, ## *p* < 0.01, and ### *p* < 0.001.

**Figure 8 pharmaceutics-15-01272-f008:**
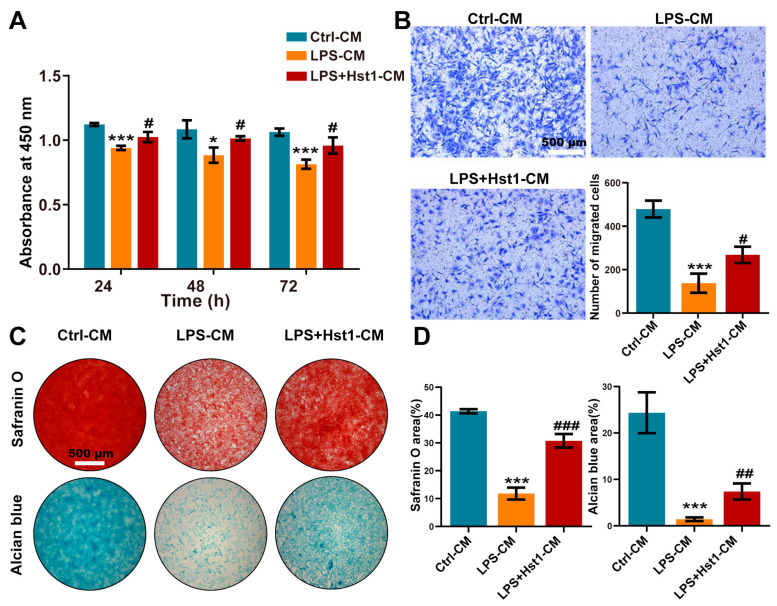
CM from Hst1-treated macrophages promoted ATDC5 cells’ metabolic activity, migration, and differentiation. (**A**) CCK8 assay was used to detect the effect of macrophage CM on the metabolic activity of ATDC5 cells. (**B**) Transwell was used to observe ATDC5 cells migration, scale bar = 500 μm. (**C**,**D**) Safranin O and Alcian blue staining of ATDC5 cells cultured with macrophage CM for 7 days and quantification, scale bar = 500 μm. Data are expressed as mean ± SD from three independent experiments. Significant differences compared to the control-CM group: **p* < 0.05 and ****p* < 0.001, and compared to the LPS-CM group: #*p* < 0.05, ##*p* < 0.01, and ###*p* < 0.001.

**Figure 9 pharmaceutics-15-01272-f009:**
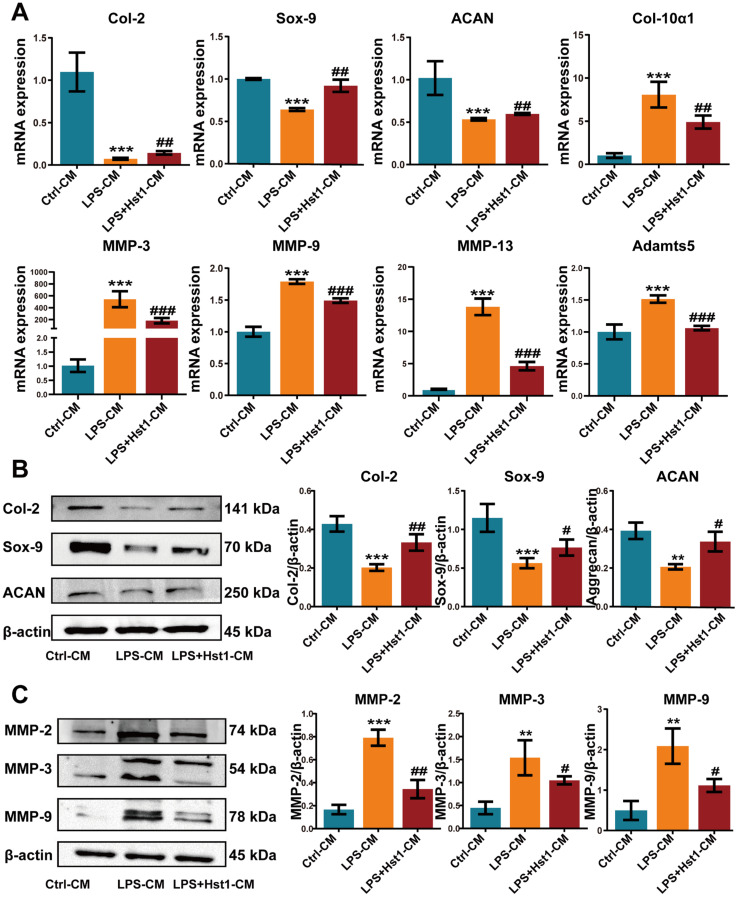
CM from Hst1-treated macrophages promoted ATDC5 cell differentiation and reduced chondrocyte destruction. (**A**) RT-qPCR results of chondrogenic differentiation markers Col-2, Sox-9, Aggrecan, and cartilage catabolism-related markers MMP-3, MMP-9, MMP-13, and Adamts5. (**B**) Western blot analysis and quantification of chondrogenic markers Col-2, Sox-9, and Aggrecan, and cartilage catabolism-related markers MMP-2, MMP-3, and MMP-9. (**C**) ATDC5 cells were cultured with macrophage CM for 7 days. Blots are cropped from Appendix A. Data are expressed as means ± SD from three independent experiments. Significant differences compared to the control-CM group: ** *p* < 0.01 and *** *p* < 0.001, and compared to the LPS-CM group: # *p* < 0.05, ## *p* < 0.01, and ### *p* < 0.001.

**Figure 10 pharmaceutics-15-01272-f010:**
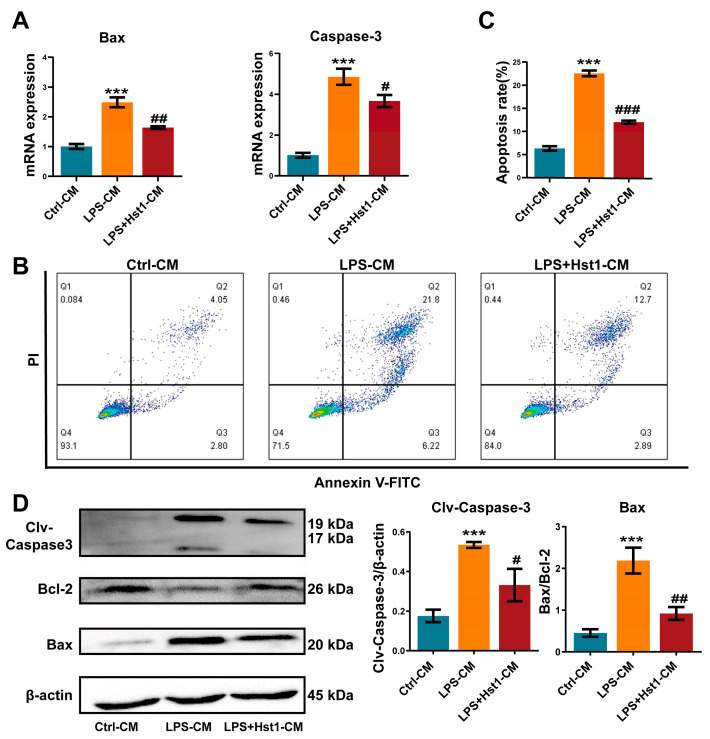
CM from Hst1-treated macrophages reduced ATDC5 cell apoptosis. (**A**) RT-qPCR analysis for apoptosis-related genes Bax and caspase-3. (**B-C**) The apoptosis rate of ATDC5 cells was analyzed using flow cytometry with annexin V-FITC/PI apoptosis analysis. (**D**) Western blot analysis and quantification of apoptosis-related proteins Bax, Bcl-2, and cleaved caspase-3. ATDC5 cells were cultured with macrophage CM for 7 days. Blots are cropped from Appendix A. Data are expressed as means ± SD from three independent experiments. Significant differences compared to control-CM group: *** *p* < 0.001, and compared to LPS-CM group: # *p* < 0.05, ## *p* < 0.01, and ### *p* < 0.001.

**Table 1 pharmaceutics-15-01272-t001:** Primers used for RT-qPCR.

Gene	Forward Primer (5′-3′)	Reverse Primer (5′-3′)
GAPDH	TGTGTCCGTCGTGGATCTG	TTGCTGTTGAAGTCGCAGGA
IL-1β	AGTGTGGATCCCAAGCAATACCCA	TGTCCTGACCACTGTTGTTTCCCA
IL-6	TAGTCCTTCCTACCCCAATTTCC	TTGGTCCTTAGCCACTCCTTC
TNF-α	CATCTTCTCAAAATTCGAGTGACAA	TGGGAGTAGACAAGGTACAACCC
iNOS	CAGAAGTGCAAAGTCTCAGACAT	GTCATCTTGTATTGTTGGGCT
CD86	CTGCTCATCATTGTATGTCAC	ACTGCCTTCACTCTGCATTTG
CD206	AGACGAAATCCCTGCTACTG	CACCCATTCGAAGGCATTC
Arg-1	CGAGGAGGGGTAGAGAAAG	CATCAACAAAGGCCAGGT
IL-10	GAGAAGCATGGCCCAGAAATC	GAGAAATCGATGACAGCGCC
Col-2	GGACGTTAGCGGTGTTGGGAG	ACTGGTGGAGCAGCAAGAGCA
Aggrecan	CAGAACCTTCGCTCCAATGAC	CCTCAATGCCATGCATCACTT
Sox-9	CAGCCCCTTCAACCTTCCTC	TGATGGTCAGCGTAGTCGTATT
MMP-3	ATGATGAACGATGGACAGATGA	CATTGGCTGAGTGAAAGAGACC
MMP-9	CACCGGCTAAACCACCTC	CGCCCGACACACAGTAAG
MMP-13	GGGGAGCCACAGATGAG	AACGCTCGCAGTGAAAAG
Admats-5	TATGACAAGTGCGGAGTATG	TTCAGGGCTAAATAGGCAGT
BAX	TGGTTGCCCTCTTCTACTTTG	GTCACTGTCTGCCATGTGGG
Caspase-3	CTGACTGGAAAGCCGAAACTC	CGACCCGTCCTTTGAATTTCT

**Table 2 pharmaceutics-15-01272-t002:** Antibodies used for Western blot (WB) and immunofluorescence (IF).

Antibody	Application and Dilution	Manufacturer
Anti-rabbit IgG	1:3000 for WB	Cell Signaling Technology, USA
Anti-mouse IgG	1:3000 for WB	Cell Signaling Technology, USA
Rabbit anti-β-actin	1:3000 for WB	Cell Signaling Technology, USA
Rabbit anti-CD86	1:1000 for WB	Cell Signaling Technology, USA
Rabbit anti-CD206	1:1000 for WB; 1:500 for IF	Abcam, UK
Rabbit anti-iNOS	1:500 for IF	Cell Signaling Technology, USA
Rabbit anti-NF-κB	1:1000 for WB	Cell Signaling Technology, USA
Rabbit anti-IKB-α	1:1000 for WB	Cell Signaling Technology, USA
Rabbit anti-pNF-κB	1:1000 for WB	Cell Signaling Technology, USA
Rabbit anti-pIKB-α	1:1000 for WB	Cell Signaling Technology, USA
Rabbit anti-P38	1:1000 for WB	Cell Signaling Technology, USA
Rabbit anti-Erk	1:1000 for WB	Cell Signaling Technology, USA
Rabbit anti-JNK	1:1000 for WB	Cell Signaling Technology, USA
Rabbit anti-pP38	1:1000 for WB	Cell Signaling Technology, USA
Rabbit anti-pErk	1:1000 for WB	Cell Signaling Technology, USA
Rabbit anti-pJNK	1:1000 for WB	Cell Signaling Technology, USA
Rabbit anti-Col-2	1:1000 for WB	Cell Signaling Technology, USA
Rabbit anti-Sox-9	1:1000 for WB	Abcam, UK
Mouse anti-Aggrecan	1:1000 for WB	Cell Signaling Technology, USA
Rabbit anti-MMP-2	1:1000 for WB	Abcam, UK
Rabbit anti-MMP-3	1:1000 for WB	Abcam, UK
Rabbit anti-MMP-9	1:1000 for WB	Abcam, UK
Rabbit anti-Clv-Caspase-3	1:1000 for WB	Cell Signaling Technology, USA
Rabbit anti-Bcl-2	1:1000 for WB	Cell Signaling Technology, USA
Rabbit anti-Bax	1:1000 for WB	Cell Signaling Technology, USA

## Data Availability

The data and material that support the findings of this study are available from the corresponding author upon reasonable request.

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
