# Peer review of "Human Salivary Histatin-1 Attenuates Osteoarthritis through Promoting M1/M2 Macrophage Transition"

_pharmaceutics, 2023, doi:10.3390/pharmaceutics15041272_

Round 1

Reviewer 1 Report

The manuscript by Wu A et al investigated the efficacy of Hst1 in attenuating articular tissue damage in OA. The study is potentially interesting since it focuses on an important topic in the field of OA but its strength is reduced by the fact that caution should be made in interpreting the present results and the data are not sufficient to support some of the Authors proposed conclusions. Moreover, a lot of inaccuracies and oversights are present in the paper and some of the references should be revised. For this reasons, partial re-writing and reorganization of the manuscript is required and English language needs revision.

1. In the Abstract section there are a lot of inaccuracies that should be corrected:

- Line 25, please define “MIA”

- Lines 33, please change “...Cell migration...” with “…cell migration…”

- Lines 37-44, please delete the paragraph “…single paragraph of about 200 words maximum.

For research articles, abstracts should give a pertinent overview of the work. We strongly encourage authors …. the main conclusions “.

2. In the Introduction section reference no. 2 should be replaced with a more suitable one.

3. In the “Materials and Methods” a lot of oversights are present and the section should be revise.

4. Results section:

a) Please, indicate if the results obtained with cell cultures are presented as mean of three independent experiments.

b) In my opinion, too many Figures have been inserted in the main text. Some of them could be moved in the supplementary file.

5. Figure legends: please reformulate the figure legends to help readers’ understanding, adding more details and descriptions and providing information about the test used for the statistical analysis.

6. The Discussion section must be improved by also adding considerations/limits of the study.

Author Response

Comment 1. In the Abstract section there are a lot of inaccuracies that should be corrected:

  1. Line 25, please define “MIA”
  2. Lines 33, please change “...Cell migration...” with “…cell migration…”
  3. Lines 37-44, please delete the paragraph “…single paragraph of about 200 words maximum.
  4. For research articles, abstracts should give a pertinent overview of the work. We strongly encourage authors …. the main conclusions “.

Reply: We sincerely thank the reviewer for the pertinent advice and we have modified the Abstract as requested.

Comment 2. In the Introduction section reference no. 2 should be replaced with a more suitable one.

Reply: We added the relevant reference.

Comment 3. In the “Materials and Methods” a lot of oversights are present and the section should be revise.

Reply: We sincerely thank the reviewer for the pertinent advice and we have modified it.

Comment 4. Results section:

  1. Please, indicate if the results obtained with cell cultures are presented as mean of three independent experiments.
  2. In my opinion, too many Figures have been inserted in the main text. Some of them could be moved in the supplementary file.

Reply: (a) We provided this information in each figure legend. (b) we have replaced Figure 3 in supplementary Figure1.

Comment 5. Figure legends: please reformulate the figure legends to help readers’ understanding, adding more details and descriptions and providing information about the test used for the statistical analysis.

Reply: We rewrote the Figure legends. The statitical test used has been mentioned in statistical analysis part.

Comment 6. The Discussion section must be improved by also adding considerations/limits of the study.

Reply: We added the following text in the last paragraph of the discussion section: “Although, this study extensively analyzed the efficacy of Hst1 in inflammation modulation-mediated attenuation of bone and cartilage damage in OA, some limitations do exist. The mechanism of Hist-1-mediated effect on macrophage immune modulation should be further studied mainly focusing on Hist-1 receptors in macrophages. Moreover, the possible systemic adverse effect of Hist-1 treatment should be thoroughly investigated during in vivo study to evaluate the biological safety.”

Reviewer 2 Report

Major

This is a descriptive study. There is no attempt to determine the most proximal molecular target of histatin-1.

There are 6 histatins in humans. The rationale for the study should have been to examine the effects of 6 peptides in the M1 to M2 macrophage phenotypic transition assay and choose the most effective peptide from the results.

Minor

Sentences from lines 37-45 are related to the author’s instruction. Please delete them.

For the dose of histatin-1 used, references are required (line 108). The duration and regimen of the treatment should be mentioned here.

The contralateral knee of the rat treated with the vehicle should act as the control. Has this been done?

Fig 3 is not necessary. Just mention it in the text as data not shown.

Fig 8, the time at which each phopho protein was studied should be mentioned in the legend. Please indicate the mol wt of each band. Same for Fig 10, B & C; Fig 11 D.

For Fig 11D, the duration of treatments should be mentioned in the legend.

Author Response

Comment 1. Sentences from lines 37-45 are related to the author’s instruction. Please delete them.

Reply: We sincerely thank the reviewer for the pertinent advice and we have removed it as requested.

Comment 2. For the dose of histatin-1 used, references are required (line 108). The duration and regimen of the treatment should be mentioned here.

Reply: We added citations related to Hist-1 dosage and treatment methods.

Comment 3. The contralateral knee of the rat treated with the vehicle should act as the control. Has this been done?

Reply: We agree with the reviewer’s concern regarding the contralater knee of the rat treated with the vehicle. Unfortunately, in this study we did not perform this part of the experiment. We will be careful about this in our future study.

Comment 4. Fig 3 is not necessary. Just mention it in the text as data not shown.

Reply: We placed Figure 3 in supplementary Figure 1.

Comment 5. Fig 8, the time at which each phosphoprotein was studied should be mentioned in the legend. Please indicate the mol wt of each band. Same for Fig 10, B & C; Fig 11 D.

Reply: We modified it accordingly.

Comment 6. For Fig 11D, the duration of treatments should be mentioned in the legend.

Reply: We mentioned the relevant information in the Figure legend of new figure 10D.

Reviewer 3 Report

In this study, for the first time, the authors demonstrated that Hst1 alleviated rat knee OA via M1 to M2 macrophage phenotype switching-mediated mitigation of inflammation and support of chondrogenic cells’ functions, suggesting the possible therapeutic potential of Hst1 to treat knee OA.

The study appears interesting, well described and argued. Minor revisions are required as follows.

Please remove from line 37 to line 45 of abstract section.

Please put apex where the number of cells is reported.

Line 161: please briefly describe the method to evaluate the cell metabolic activity or cite a paper in which this procedure is well described.

Line 172: Please switch the number of rpm with gravity (g).

Line 271: What do the authors mean by “subchondral bone”? May be “subchondral bone exposure? Please specify.

Line 518: please change “… in higher..” with “…is higher…”

Author Response

Comment 1. Please remove from line 37 to line 45 of abstract section.

Reply: We sincerely thank the reviewer for the pertinent advice and we have removed it as requested.

Comment 2. Please put apex where the number of cells is reported.

Reply: We apologize for this mistake. We modified it as requested.

Comment 3. Line 161: please briefly describe the method to evaluate the cell metabolic activity or cite a paper in which this procedure is well described.

Reply: We added relevant citations.

Comment 4. Line 172: Please switch the number of rpm with gravity (g).

Reply: We corrected it accordingly.

Comment 5. Line 271: What do the authors mean by “subchondral bone”? May be “subchondral bone exposure? Please specify.

Reply: Bone below the articular cartilage.

Comment 6. Line 518: please change “… in higher…” with “…is higher…”

Reply: We modified it as requested.

Round 2

Reviewer 1 Report

A numbers of points and concerns were addressed adequately in this second version. 

Author Response

We sincerely thank the reviewers for their time and effort.

Reviewer 2 Report

The authors have addressed my concerns satisfactorily.

Author Response

(The authors gave the same response as above.)
